# Program Language Translation Using a Grammar-Driven Tree-to-Tree Model

## Abstract

The task of translating between programming languages differs from the challenge of translating natural languages in that programming languages are designed with a far more rigid set of structural and grammatical rules. Previous work has used a tree-to-tree encoder/decoder model to take advantage of the inherent tree structure of programs during translation. Neural decoders, however, by default do not exploit known grammar rules of the target language. In this paper, we describe a tree decoder that leverages knowledge of a language's grammar rules to exclusively generate syntactically correct programs. We find that this grammar-based tree-to-tree model outperforms the state of the art tree-to-tree model in translating between two programming languages on a previously used synthetic task.

## 1. Introduction

Program translation is the process of converting code in one programming language to code in another, ideally with minimal human effort. It has the possibility to significantly alter the ways in which programs are developed. With a perfect translator, a programmer could freely choose a programming language to use without regard to whether the chosen language is the most efficient for the task at hand. Effective program translation would thus enable programmers to focus on the content and development of a specific program as opposed to the details of a particular language. With such a translation method, a developer could easily import code to different platforms, streamlining the development process.

Programming languages are similar to natural languages in many ways, and natural language translation has been studied extensively. Sequence-to-sequence translation models, which map input sequences to output sequences, have achieved great performance (Bahdanau et al., 2014; Cho

[1]Anonymous Institution, Anonymous City, Anonymous Region, Anonymous Country. Correspondence to: Anonymous Author <anon.email@domain.com>.

Preliminary work. Under review by the International Conference on Machine Learning (ICML). Do not distribute.

et al., 2014; Eriguchi et al., 2016; Vaswani et al., 2017). While similar to natural languages, programming languages have a distinct structure which makes it harder to use the same tools for translation. For instance, the RNN-based sequence generator, which easily generates phrases in a natural language, finds it difficult to generate long syntactically correct programs (Karpathy et al., 2015).

Techniques for program language translation are often variants of statistical machine translation (SMT) (Lopez, 2008), which involves modeling the probability distribution of phrases in the target language given phrases in the source language. Nguyen found SMT methods from NLP applied to programming language translation produced many syntactically incorrect programs (Nguyen et al., 2013). They later found that SMT methods could be improved by incorporating knowledge of program syntax (2016a). They also saw success matching tokens in different languages through similarities in their usage in context (Nguyen et al., 2016b).

Recently, there has been a rise in the use of neural networks for programming language tasks. Neural networks have been applied to code-generation tasks converting images to code (Beltramelli, 2017) and converting text to code (Yin & Neubig, 2017). They have also been applied to tasks like program induction (Bunel et al., 2016) and program classification (Peng et al., 2015).

Recent work applied tree-based neural networks to programming language translation (Chen et al., 2018). Their tree-to-tree encoder/decoder model performed better than sequence-to-sequence models and improved on state-of the art program translation approaches by margins of 20 points for real-world translation projects.

However, this tree-to-tree program translation model faces various issues, including the generation of syntactically invalid programs and inefficiencies stemming from the need to generate end of tree tokens at each branch of the underlying abstract syntax tree (AST). Our work makes this tree-to-tree model more efficient by leveraging the grammar of the language to generate only syntactically correct programs and removing redundant notation such as end of tree node to decrease the number of required operations in the model.

The remainder of this paper is organized as follows. Section 2 presents the tree-to-tree model and discusses prior work.

Section 3 presents the framework for our work. Section 4 describes our experiments and presents our results. Lastly, Section 5 concludes and mentions directions for future work.

## 2. Background

Our model is heavily inspired by the tree-to-tree encoder/decoder model introduced by Chen et al. Their model employs a tree LSTM (Tai et al., 2015) to encode the source tree. First, the input program tree is binarized using the Left-Child Right-Sibling representation. The input tree is then encoded by an LSTM beginning at the leaves of the tree (Chen et al., 2018).

Each node $N$ in the tree has a token $t_s$ and up to two children; a left child $N_L$ and a right child $N_R$. If the children maintain LSTM states $(h_L, c_L)$ and $(h_R, c_R)$, then the LSTM state $(h, c)$ for $N$ is computed as:

$$(h, c) = LSTM(([h_L : h_R], [c_L, c_R]), t_s). \quad (1)$$

The hidden state and cell state of any missing children are represented as vectors of zeros.

The decoder then generates the target tree by inserting the LSTM state of the root into a queue of nodes to be expanded. While the queue contains elements, one is popped out and an attention mechanism is applied to determine what nodes in the input tree are most relevant (Chen et al., 2018). The attention mechanism is based on computing a dot product of a representation of the hidden state with all the encoded representations from the input tree (Luong et al., 2015) and produces a context vector $e_s$. From there, the hidden state and the context vector are used to determine the probabilities of the next token as shown in Equations (2) and (3).

$$e_t = \tanh(W_1[e_s; h]) \quad (2)$$

$$t_t = \text{argmax softmax}(We_t) \quad (3)$$

$W$ and $W_1$ are trainable weigh matrices. If the generated token $t_t$ is not <EOS>, the decoder will then generate two children for the expanding node.

Chen et al. then generate the LSTM state for each of the node's children with another set of LSTMs $LSTM_1 \ldots LSTM_m$, where $m$ is the maximum number of children a node can have (in their case two, since output trees were binarized as well). Hidden and cell states for the $i^{th}$ child of $N$ are generated from its hidden and cell states $(h, c)$ as follows:

$$(h_i, c_i) = LSTM_i((h, c), [Bt_t; e_t]) \quad (4)$$

where $B$ is an embedding matrix. To help the LSTM incorporate information from a node's attention when generating its children, they use parent attention feeding — concatenating the embedding representation of the parent's value with its attention vector before feeding them into the LSTM. The child nodes are then pushed into the queue of nodes to be expanded. Tree generation stops when the queue is empty.

Recent work by Yin and Neubig (2017) developed a grammar-based neural architecture. Their neural network was able to generate complex Python programs from natural language descriptions by converting a natural language statement into a syntactically correct AST for the target language. This network, however, generates nodes as a series of sequential instructions to extend or terminate a tree branch rather than directly utilizing the tree structure.

In this paper, we apply the concept of a grammar model to the task of program language translation. We leverage the target language's grammar rules to enhance translation accuracy. The benefits of using grammar rules are that they generate only syntactically valid programs and they eliminate the redundancy of the end of tree token, thus increasing training speed.

## 3. Grammar-Based Tree-to-Tree model

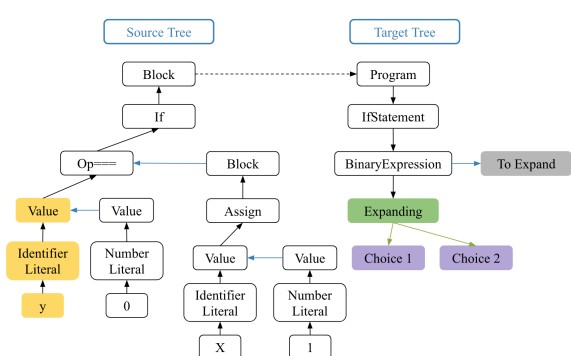

*Figure 1.* This diagram, adapted from Chen et al., shows a target tree being generated by a binarized input tree. When a node is expanded (green block), it can only generate children from grammatically correct choices (purple blocks) and add them to the queue of unexpanded nodes (gray). Our attention mechanism lets the decoder focus on relevant nodes in the input (yellow blocks).

We implement a tree-to-tree encoder/decoder model patterned off (Chen et al., 2018). The tree encoder is almost identical to the one described in the paper except that our model does not use dropout (Srivastava et al., 2014) as we did not find it improved training accuracy. However, we modify their tree decoder to make use of the grammar of the target language when generating nodes. Unlike in the paper

by Chen et al., we do not binarize the output tree, because the target language's grammar rules cannot easily be applied to trees in which a node's siblings can appear as its children.

To generate a node's children, we first call a function that returns a list of the categories of tokens that can be generated by that node at each child index. A category is a unique set of valid children. For instance, in the language FOR (described further in the Experiments section), the node $<$PLUS$>$ can generate tokens in the $Expression$ category: ($<$PLUS$>$, $<$MINUS$>$, $<$VAR$>$, or $<$CONST$>$). Each category $k$ has an associated learnable weight matrix $\boldsymbol{W}_k$.

To generate each child, we create a node embedding $e_t$ computed the same way as the tree decoder by (Chen et al., 2018). We then generate the node's token by finding the most probable token out of the set of possibilities as follows:

$$t_t = \textbf{argmax} \ \textbf{softmax}(W_k e_t). \tag{5}$$

Note that as the number of tokens in class $k$ is substantially fewer than the total number of tokens in the language, this is an easier prediction problem than for the prior tree decoder.

After this, as in (Chen et al., 2018), we feed an embedded representation of $t_t$ into an LSTM to compute the hidden and cell state for each of its children. We always train using teacher forcing (passing the true value of $t_t$ into the LSTM rather than the generated value) because a single incorrect token that generates different categories of children than the correct token could make the probability of generating a correct token for any of its children zero. This could zero out most of our gradients, slowing down training.

The decoder iteratively creates nodes, starting at the root and generating each child node from the hidden and cell states generated for that child from its parent. Since the program's grammar does not allow tokens to be generated from terminal tokens, branches of the tree end automatically when a terminal is produced. This means our grammar decoder does not have to learn to generate $<$EOS$>$ tokens, simplifying the translation task and decreasing the number of operations the model needs to perform.

## 4. Experiments

We tested our grammar-based tree-to-tree model on a task described by Chen et al. which examines the ability of a model to translate between simple programming languages of different paradigms. For the task, we randomly generated a synthetic dataset of 100,000 training programs in FOR, a simple imperative programming language created by (Chen et al., 2018). We also generated test and validation sets with 10,000 programs. The programs were generated by an almost context-free probabilistic grammar. It is not fully context-free since to avoid generating programs that used

*Table 1.* FOR/LAMBDA training dataset description.

| METRIC | FOR | LAMBDA |
|---|---|---|
| TOTAL PROGRAM COUNT | 100K | 100K |
| AVERAGE PROGRAM LENGTH | 22 | 56 |
| MINIMUM PROGRAM LENGTH | 5 | 13 |
| MAXIMUM PROGRAM LENGTH | 104 | 299 |
| NUMBER OF TOKENS IN LANGUAGE | 32 | 33 |

variables before they were defined, we kept track of previously defined variables and only used those in expressions. These programs were then fed into a translator function that converts them into a simple functional language called LAMBDA also created by (Chen et al., 2018). More dataset details are available in Table 1.

In Table 2, we compare our model's performance to the baselines described by (Chen et al., 2018) and to our own reimplemented tree-to-tree and tree-to-sequence models with the architecture and hyperparameters described by Chen et. al. For the hyperparameters of the grammar model we simply used the tree-to-tree model's hyperparameters and did not try to optimize them. Each model was trained 5 times over half a million examples. We also ran one of each of our three models to convergence (30 epochs). Program accuracy was measured on the test set by counting the percentage of perfectly correct translated programs.

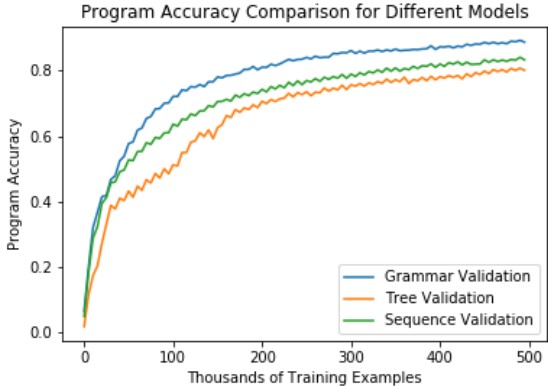

*Figure 2.* Average validation accuracy for grammar, tree2tree, and tree2seq model for 5 epoch runs.

Results are summarized in Figure 2 and Table 2. The grammar-based model achieved an average 88.82% accuracy, outperforming our reimplementation of the tree-to-tree model and tree-to-seq model. When run to convergence, the difference in accuracy among the models decreased but the grammar model remains more accurate. In addition, the grammar model converges more stably than the other models as the standard deviation of its accuracy is much

*Table 2.* Program Accuracy on the FOR/LAMBDA translation task. Since our datasets are not identical, performance of the reimplemented Tree2Tree and Tree2Seq models does not perfectly match the results reported by (Chen et al., 2018). The Chen et al. have some entries as N/A as they don't mention multiple runs. The last three rows display the results of running our models to complete convergence.

| MODEL | MEAN ACCURACY | σ ACCURACY |
| --- | --- | --- |
| **GRAMMAR TREE2TREE** | 88.82% | 0.64% |
| REIMPLEMENTED TREE2TREE | 80.69% | 7.02% |
| REIMPLEMENTED TREE2SEQ | 83.59% | 3.95% |
| CHEN ET AL. TREE2TREE (EASY) | 99.76% | N/A |
| CHEN ET AL. TREE2SEQ (EASY) | 98.36% | N/A |
| CHEN ET AL. TREE2TREE (HARD) | 97.50% | N/A |
| CHEN ET AL. TREE2SEQ (HARD) | 87.84% | N/A |
| **(LONG) GRAMMAR TREE2TREE** | 93.70% | N/A |
| (LONG) REIMPLEMENTED TREE2TREE | 91.89% | N/A |
| (LONG) REIMPLEMENTED TREE2SEQ | 90.60% | N/A |

smaller (while the tree2tree model's convergence depends a lot more on random seed). The grammar model's average run beat the best run of both baselines.

The reimplemented tree-to-tree and tree-to-seq models performed worse than the results reported by (Chen et al., 2018). This discrepancy could be caused by differences between the complexity of our datasets. Since their dataset was unavailable, we implemented a dataset with similar average program lengths. However, our dataset may have used more variables or constants, and it had much greater variation in program lengths. Our average program length was about the length of the programs in the easy version of their task, but our programs with longest length were about twice as long as the programs in the hard version of their task. It is also possible that despite our attempt to faithfully re-implement (Chen et al., 2018) that there are slight differences between the two tree-to-tree models as we lacked their code. Any of those differences would be shared with the grammar model as except for the code corresponding to the decoder, they shared all of their code. To make it easier for future research to evaluate on the same task, we release our dataset and experiment code here, `https://www.dropbox.com/sh/1q4aejr57jk40fs/AADKTvgKqLHuIIzNdlANjGRea?dl=0`.

## 5. Conclusion

This paper proposes a grammar-based program language translation approach using a grammar decoder that outperforms the state of the art tree-to-tree models for program language translation in both number of operations needed and accuracy. Future work will explore ways to improve convergence and broaden the practical applicability of our approach to various real programming translation problems.

One limitation to this approach is the practical difficulty of obtaining the training data needed to apply this model to a new pair of languages. Training requires a parallel corpus of programs in two languages. Previous researchers such as (Chen et al., 2018) have obtained such datasets by using languages with an explicit translator between them (which largely obviates the need for a neural translation program) or by finding real-world programs implemented in both programs (which may be difficult to obtain).

Our model also requires a formal grammar for the target language. In the absence of a grammar, we could approximate a grammar from the training set by recording all child tokens of each unique token, but this could make our model incapable of generating rare but valid syntactic patterns.

Finally, our model currently caps the number of variable names and literals at a fixed number determined before training. Future work could explore alternative ways to generate arbitrarily many variables and literals by copying them from the input program using a method similar to that implemented by (Yin & Neubig, 2017).

In future work, we will integrate other ideas from natural language translation to our tasks. One possibility includes self-attention (Vaswani et al., 2017), a mechanism that provides the model at each time step with its state at previous time steps and may help the model learn more complex relationships among parts of the program. We could also integrate a language model into our decoder to help with translating unusual expressions not seen in the training data.

Our current approach makes parallelization of the training process difficult. Since every tree has a different structure, we cannot batch training examples together for faster processing on GPUs. Sequence-to-sequence models can circumvent this by batching programs of the same size or padding shorter sequences. We will need to explore methods of batching tree generation and apply them to our model.

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
