# OpenReview forum: "Program Language Translation Using a Grammar-Driven Tree-to-Tree Model"
_ICML.cc/2018/Workshop/NAMPI — NAMPI 2018_

### Review · AnonReviewer3 · 2018-06-26
**Nice experimental paper that would benefit from participation in workshop**

**Rating:** 7
**Confidence:** 4

**Review:**

This paper presents an encoder-decoder (tree-to-tree) approach for mapping code in one programming language to another programming language. The key innovation is a new tree-based decoder, which better represents the hierarchical nature of the output code. Experiments demonstrate that the approach outperforms the tree-to-tree model of Chen et al 2018.

Overall, this paper presents a nice contribution and I think the authors would benefit from attending the workshop to present the work. I had a few points where things could possibly be improved:
* The details of the decoder in paper are a little hard to follow, and could possibly benefit from a more formal / mathematical presentation. The key idea seems to be to do top down propagation of the embedding, instead of left to right at each level of the grammar. This is nice, if so, and it is interesting to see it makes such a big difference. You might consider compressing the background and discussion to make room for a more detailed presentation of your approach.
* The evaluation data seems good, but it is too bad you couldn't also compare of previous datasets. Was there a technical reason for this restriction? Chen et al 2018 seem to use challenging benchmarks? You say their data isn't available, but using real-world programs of some sort would greatly improve the work.
* I don't understand why you need so many different accuracy metrics in Table 2. Could you just report accuracy, and simplify things significantly?

---

> ### Comment · ~Olivia_Watkins1 · 2018-07-11
> **Responses to comments**
>
> Thanks for your taking the time to review our paper.  I’d like to clarify a few points in response to your comments:
> 1) The key change we’re proposing is actually to constrain tree-to-tree models to generate programs consistent with the target language’s grammar.
> 2) Chen et. al’s constructed datasets (For to Lambda and Javascript to CoffeeScript) weren’t available.  Shortly before the paper was due we did get access to a raw version of the Java to C# dataset, but we didn’t have time to train these models and include their results.
> 3) We reported the mean and standard deviation of our accuracy to show that the grammar model shows less variation.

---

### Review · AnonReviewer2 · 2018-06-28
**Simple extension to tree-decoders, underwhelming results**

**Rating:** 5
**Confidence:** 4

**Review:**

This paper considers the task of program language translation using neural encoder/decoder models. Given that existing tree-based decoders often generate programs with syntax errors, this paper extends the decoder to be grammar aware such that only valid trees in the target programming language are generated. Experiments on a dataset shows that the proposed model results in more accurate code translation.

I think the paper is quite well-written and clear, and it is clear that the decoder should only produce syntactically correct code, thus is well-motivated. However, similar ideas have been explored using LSTMs to generate trees in the NLP literature, and the results were underwhelming, making me hesitant to give a acceptance recommendation.


To elaborate, there has been some work in neural semantic parsing in incorporating grammar production rules when generating a tree at decoding time, see https://aclanthology.info/papers/D17-1160/d17-1160 for example. These are similar to the Yin and Neubig paper that the authors cite, but decode a linearized representation of the tree. Thus, I found the contribution of the paper to be quite incremental.

Secondly, I was expecting the results to be much better, but it seems existing sequence and tree decoders actually capture the grammar quite well. Maybe if the authors had provided more analysis, such as breakdown of the accuracy by the length of the trees, etc. the utility of the proposed approach would have been more clear.

---

### Decision · ~NAMPI_Admin1 · 2018-06-28
**Paper8 Final Decision**

Accept